# Patient Satisfaction with the Expanded Nurses Service in Primary Health Care: Evidence from Kazakhstan

**DOI:** 10.3390/healthcare13243314

**Published:** 2025-12-18

**Authors:** Indira Abdikadirova, Lyudmila Yermukhanova, Aurelija Blaževičiene, Zhanar Dostanova, Zaure Baigozhina, Maiya Taushanova, Gulnar Sultanova, Kauysheva Almagul

**Affiliations:** 1Department of Public Health and Health Care, West Kazakhstan Marat Ospanov Medical University, Aktobe 030019, Kazakhstan; abdikadirovait@gmail.com (I.A.); yermukhanova@zkmu.kz (L.Y.); dr.taushanova@zkmu.kz (M.T.); g.sultanova@zkmu.kz (G.S.); 2Department of Nursing, Lithuanian University of Health Sciences, 44307 Kaunas, Lithuania; aurelija.blazeviciene@lsmu.lt; 3School of Nursing, Astana Medical University, Astana 010000, Kazakhstan; 4National Scientific Center of Phthisiopulmonology of the Republic of Kazakhstan, Almaty 050000, Kazakhstan; aleke.astana@gmail.com

**Keywords:** advanced practice nursing, primary health care, patient satisfaction, nurse practitioner

## Abstract

Background/Objectives: The implementation of advanced practice nursing in Kazakhstan is aimed at improving the accessibility and quality of primary healthcare. One of the key indicators of the effectiveness of this model is patient satisfaction, which reflects the perceived quality of care and directly influences treatment adherence. The aim of the study was to assess patient satisfaction with nurse-led consultations in primary healthcare institutions in Kazakhstan. Methods: A cross-sectional study was conducted using a questionnaire developed on the basis of Karin Bergman’s instrument and adapted to the Kazakhstani context. A total of 621 patients who attended independent nursing consultations in polyclinics in Aktobe, Almaty, Astana, and the village of Merke participated in the survey. Descriptive statistics and Pearson’s χ^2^ test were applied, with statistical significance set at *p* < 0.05. Results: The majority of respondents were women, with a median age of 61 years. Awareness of independent consultations was higher among patients who regularly visited nurses (97.1% vs. 86.9%; *p* < 0.006). High satisfaction levels were associated with service accessibility, quality of examination, and clarity of recommendations. Among regular visitors, 99.2% reported satisfaction with the nurse’s work, and 76.6% rated the service as “excellent”. In contrast, patients with irregular visits more often reported dissatisfaction due to insufficient attention and limited knowledge of nurses. Conclusions: The findings confirm a high level of patient satisfaction with advanced practice nursing services and highlight the importance of this model in strengthening primary healthcare in Kazakhstan.

## 1. Introduction

Strengthening primary health care is recognized as one of the key directions of healthcare reform in the Republic of Kazakhstan [1,2]. In the context of population aging, the growing prevalence of chronic non-communicable diseases [3,4,5], a shortage of physicians, and the need to improve the accessibility and quality of outpatient services, the introduction of the advanced practice nursing model has acquired particular importance [6,7,8,9].

An advanced practice nurse, equipped with competencies that enable the performance of extended clinical functions—including diagnosis, prescribing of pharmacological treatment, and patient management within defined settings—emerges as an essential resource for optimizing healthcare delivery at the primary care level.

International experience from countries with highly developed healthcare systems, such as the United States, Canada, the United Kingdom, and Australia, confirms the effectiveness of advanced nursing practice as a means of improving patient satisfaction, expanding service coverage, and reducing the burden on physicians. In these countries, nurses with clinical decision-making autonomy are actively involved in managing patients with chronic diseases, providing preventive care and counseling, thereby ensuring a high level of public trust [10].

In addition to international data on nurse-led care, there are several established frameworks and instruments that health systems use to systematically assess patient experience, satisfaction, and perceived quality of care. For example, the Patient Satisfaction Questionnaire (PSQ) is widely used to evaluate multiple dimensions of care, including provider communication, technical competence, and accessibility [11]. The Picker Patient Experience Questionnaire (PPE-15) focuses on patient-centered care and experience across multiple domains, such as respect for patient preferences, emotional support, and coordination of care [12]. In the United States, the Consumer Assessment of Healthcare Providers and Systems (CAHPS) surveys provide standardized assessment of patient experience with providers and health plans, widely used for quality improvement and policy decisions. Additionally, consolidated primary care performance frameworks have been applied in countries such as Canada and the UK to evaluate service delivery, patient outcomes, accessibility, equity, and integration of care. These frameworks allow researchers and health system evaluators to capture not only clinical outcomes but also organizational, communicative, and emotional dimensions of care that are critical for patient satisfaction [13].

In Kazakhstan, the practice of independent consultations conducted by nurses is still at an early stage of development. Since 2018, pilot projects have been implemented to introduce a new model of nursing services aimed at expanding professional autonomy and broadening the functional responsibilities of nurses [14]. In Kazakhstan, an advanced practice nurse is a specialist with post-secondary or higher nursing education who has completed advanced training and is authorized to perform an expanded scope of practice. These professionals provide independent nursing consultations and conduct dynamic patient monitoring, counseling, health education, and home visits within disease management programs and the universal progressive home-visiting model. The advanced practice nurse’s competencies also include health promotion and disease prevention activities, the implementation of screening programs and vaccinations, and performing a range of diagnostic and therapeutic procedures delegated by a physician.

However, one of the most important indicators of the effectiveness of this model is the opinion of patients as the direct recipients of healthcare services [15].

Patient satisfaction reflects not only the subjective perception of the quality of care provided but also objectively influences treatment adherence, compliance with medical recommendations, and ultimately, clinical outcomes. In this regard, assessing patient satisfaction with services received during independent consultations with a nurse makes it possible to identify the strengths and weaknesses of implementing advanced nursing practice, as well as to develop measures for its further improvement.

Previous studies conducted in several European countries have shown that patients’ lower evaluations were related not so much to the clinical component, but rather to organizational and communicative aspects—waiting times, interactions with medical staff over the phone, and insufficient attention to the emotional and personal needs of patients [16]. These dimensions constitute an essential part of the competencies of practicing nurses, making them a critical component in improving patient satisfaction within the primary healthcare system [17].

Despite the availability of international evidence on independent nursing consultations, Kazakhstan lacks systematic studies on patient satisfaction with independent consultations provided by advanced practice nurses. To date, little is known about which aspects of this model patients evaluate most positively or critically, as well as how they perceive the expanding role of nurses in primary health care. This gap limits the understanding of the effectiveness of ongoing reforms.

The aim of the study was to access patient satisfaction with nurse-led consultations in primary healthcare institutions in Kazakhstan.

## 2. Materials and Methods

### 2.1. Study Design

This was a cross-sectional study conducted over an eight-month period (April–November 2023).

### 2.2. Measurement Instrument

Data were collected using a specially designed questionnaire, previously employed in the study by Karin Bergman and colleagues [18], aimed at assessing patient satisfaction with healthcare services provided by advanced practice nurses in the United Kingdom. The original instrument was adapted and refined to suit the context of the healthcare system in the Republic of Kazakhstan. The instrument was registered in the State Register of Rights to Objects Protected by Copyright (No. 31620, 6 January 2023) under the title ‘Study of Patient Satisfaction with Independent Nurse Consultations in Primary Health Care’. The questionnaire was pretested to enhance its validity and reliability. The Cronbach’s alpha reliability coefficient for the questionnaire items in our study was 0.7, indicating an acceptable level of internal consistency, and modifications were made based on the results of the pilot testing. The final version of the questionnaire comprises 22 items grouped into four domains: socio-demographic characteristics, patient awareness of the nurse-led consultation, organizational aspects, and patient perceptions of the quality of the independent nurse consultation. The questionnaire includes both closed and open-ended questions. Special attention was given to clarity and accessibility in the wording of questions, with deliberate avoidance of specialized medical terminology and abbreviations, ensuring that the questionnaire was understandable to a wide range of respondents.

### 2.3. Study Setting and Participants

A total of 621 patients who attended consultations with advanced practice nurses participated in the study, representing four healthcare organizations: City Polyclinic No. 3 in Aktobe, City Polyclinic No. 5 in Almaty, City Polyclinic No. 5 in Astana, and the Merken Central District Polyclinic. Participants were recruited consecutively during their visits to the selected healthcare facilities over the eight-month study period (April–November 2023). Eligible patients were invited to participate by the research team involved in the scientific project, who were not engaged in their clinical care. The survey was administered immediately after the advanced practice nurse consultation to accurately capture patients’ perceptions and minimize recall bias; after providing verbal and written informed consent, participants completed the questionnaire, either independently or with the assistance of a researcher who did not influence their responses. All data were anonymized, and data entry and analysis were conducted by researchers not involved in data collection, ensuring objectivity and the reliability of the results.

Sample size was calculated using the Raosoft calculator assuming a 95% CI and 5% margin of error (required *n* = 381); a 20% contingency was added, yielding a final sample of 621, which further increased the reliability of the findings. Participant selection followed predefined inclusion and exclusion criteria, as summarized in Table 1.

### 2.4. Statistical Analysis

Categorical variables were described using descriptive statistics by presenting absolute (n) and relative (%) frequencies. To assess statistically significant differences between groups in categorical variables, Pearson’s chi-square test (χ^2^) was applied. In cases where the assumptions of the χ^2^ test were violated (e.g., when expected cell frequencies were less than 5), an alternative test—the likelihood ratio chi-square test—was used. Statistical significance was determined at a level of *p* < 0.05. All statistical tests were conducted two-tailed. The initial data entry was performed in the Microsoft Excel spreadsheet editor, after which the data were imported into IBM SPSS Statistics 26 software (IBM Corp., Armonk, NY, USA).

### 2.5. Ethical Considerations

This study was conducted in accordance with the principles of the Declaration of Helsinki. All patients received written information and provided signed informed consent before their study participation. The study was approved by the Bioethics Committee of West Kazakhstan Medical University named after Marat Ospanov (protocol No. 3, dated 14 March 2023).

## 3. Results

To assess satisfaction with independent nursing consultations among the registered population, patients were divided into two categories based on the frequency of visits to the nurse: “regular visitors” (at least once a month, predominantly patients with chronic diseases under dispensary supervision) and “rarely visitors” (occasional visits for preventive check-ups, medical certificates, or specific health issues, patients not under dispensary supervision).

Main characteristics of patients. According to gender distribution, the majority of respondents were women (64.6%), while men accounted for 36.1% of the total sample, indicating a significant predominance of female participants in the study group. The median age of participants was 61 years, with most falling within the age range of 51 to 68 years, reflecting a predominance of middle-aged and older adults. The largest proportion of respondents had secondary education (55.3%), whereas higher education was reported by 26.3% of participants. Secondary vocational education was noted in 12.6% of cases, and the smallest group consisted of individuals with incomplete secondary education (5.7%). With respect to marital status, the overwhelming majority of participants were married (75.7%); widowed individuals accounted for 17.7%, divorced—3.9%, and those who had never been married—2.7%. In terms of social status, the largest category was represented by retirees (43.8%), followed by workers (38%), while other categories accounted for the smallest proportion. The main socio-demographic characteristics of patients are presented in Table 2.

### 3.1. Patient Awareness of an Independent Consultations

One of the key factors influencing the successful implementation and functioning of the advanced practice nursing model within the primary health care system is the level of public awareness regarding the availability of medical services provided through independent consultations conducted by nurses. Patients were divided into two categories based on the frequency of their visits to the nurse. The first category included patients who attended nurse-led consultations regularly (at least once a month); the second included those who attended infrequently or occasionally.

Table 3 shows that among patients who regularly attend appointments, 97.1% are aware of the independent nursing consultation, whereas among those who visit less frequently, this figure is 86.9% (*p* < 0.006), indicating a statistically significantly higher level of awareness of this service among regular attendees. At the same time, 2.9% of patients in the first category are unaware of this care model, compared to 13.1% in the second category. The main source of information through which patients learn about the “Independent Nursing Consultation” is physicians (51.5% among regular attendees vs. 33% among less frequent attendees, *p* = 0.006). Meanwhile, information provided directly by nurses accounts for 50.8% and 52.5%, respectively, with no statistically significant difference observed. The role of other sources, such as relatives, friends, the call center, and registry staff, in informing patients about the consultation is minimal in both groups.

### 3.2. Organizational Aspects of Independent Nursing Consultations

The questions presented in Table 4 are organizational in nature and reflect the corresponding aspects of structuring the nursing care process. The main reason for seeking care among most participants in both groups was the receipt of subsidized medications (prescription issuance). However, this indicator was higher among patients who regularly attended consultations (88.7%) compared to those who rarely attended (52.5%). Follow-up monitoring was also more frequent in the regularly attending group (74.6% vs. 37.7%, *p* = 0.001). Preventive examinations were reported as reasons for attendance as well (60.3% vs. 29.5%, *p* = 0.001). Statistically significant differences were observed with regard to screening visits (54.8% vs. 23.0%, *p* = 0.001). According to the study results, the majority of patients who regularly attended consultations came through walk-in visits (77.7%), whereas this proportion was lower among those who rarely attended (60.7%, *p* < 0.001). Interestingly, call-backs from nurses were more frequently reported among the rarely attending group (23.0% vs. 5.9%), while the use of electronic platforms (Damumed/E-gov) was similar in both groups (16.4%).

Waiting times play an important role in patients’ perception of the quality of medical care. Most patients reported waiting 5–10 min before the consultation began. This was noted by 55.7% of regular visitors and 60.7% of infrequent visitors. The overwhelming majority of patients who regularly attended nurse consultations (99.2%) confirmed the necessity of such consultations in polyclinics, while a slightly smaller proportion was observed among infrequent visitors (95.1%). At the same time, only 0.8% of patients in the first group indicated that nurse consultations were unnecessary, whereas this figure was 4.9% among the second group, which exceeds the proportion of negative responses in the first group (*p* = 0.029) (Table 4).

### 3.3. Quality of Independent Nursing Consultations

Within the framework of this study, particular attention was paid to how comprehensively and effectively nurses perform their professional expand functions during independent consultations. Our interest extended beyond patients’ overall perception of the services provided to the actual implementation of key components of the consultation—from history taking and patient assessment to the provision of recommendations, completion of documentation, and delivery of preventive care.

According to the data presented in Table 5, when asked, “Does the nurse conduct an examination and assess your general condition?”, the majority of patients in both groups responded positively: 91.2% in the group with frequent consultations and 80.3% in the group with infrequent visits. Negative responses were reported by 4.4% and 8.2% of patients, respectively (*p* = 0.024). Patients who answered “yes” were then asked a follow-up question: “Does the nurse ask enough questions to adequately assess your general condition?” Among them, 97.3% of patients with regular consultations and 88.5% of those with infrequent consultations considered the number of questions asked to be sufficient (*p* < 0.001).

Monitoring treatment dynamics plays a crucial role, as it enables the timely identification of changes in a patient’s condition, such as side effects or health deterioration. This, in turn, allows for appropriate measures to be taken before serious complications develop. According to the survey data presented in Table 6, overall monitoring of treatment dynamics did not reveal statistically significant differences between the groups (*p* = 0.105). However, the modes of implementing this monitoring differed: with regular consultations, nurses were more likely to invite patients for follow-up visits (89.3% vs. 67.2%, *p* < 0.001), whereas in the group with infrequent visits, home visits were more commonly practiced (18.0% vs. 3.2%, *p* < 0.001).

An equally important component of the nurse’s work during independent consultations is prevention. A patient’s health largely depends on how professionally and competently the nurse provides recommendations regarding disease prevention. As shown in Table 6, 98.3% of patients who regularly attend nurse consultations reported that the recommendations on prevention and possible complications were clear, whereas among those who rarely visited, this figure was 90.2%. Moreover, in the group of infrequent visits, the proportion of patients who found the recommendations unclear or were not provided with them at all was ten times higher (9.9% vs. 1.7%) (*p* < 0.001). The data analysis indicates the presence of statistically significant differences between the groups. A total of 87.4% of patients with regular visits reported an improvement in their well-being, compared to 73.8% among those with infrequent visits. These findings emphasize the importance of regular consultations with healthcare professionals in achieving positive health outcomes. At the same time, 12.4% of patients in the first group reported no changes after nursing intervention, while this indicator was higher in the group of infrequent visitors (23.0%). Only a small proportion of patients reported deterioration: 0.2% in the first group and 3.3% in the second (*p* < 0.002).

### 3.4. Patient Satisfaction with Nurses at Independent Consultations

When discussing patient satisfaction, it is important to emphasize that this indicator is one of the key measures of the quality of healthcare services and directly reflects the effectiveness of nurses in the context of independent consultations. Table 7 presents data on patient satisfaction with nurse-led consultations, reasons for dissatisfaction (if any), and the evaluation of their performance on a five-point scale among both patient groups. Patients who regularly visit nurses for independent consultations report significantly higher satisfaction with their services: 99.2% compared to 91.8% among those who attend infrequently (*p* < 0.001). Moreover, reasons for dissatisfaction are more frequently reported in the infrequent-visit group, such as insufficient attention to the patient (6.6% vs. 0.4%) and perceived lack of knowledge (6.6% vs. 0.8%) (*p* < 0.002). The assessment of nurse performance using a five-point scale also reveals substantial differences: 75.2% of patients with regular visits rated the service as “excellent”, whereas only 60.7% of infrequent visitors did so (*p* < 0.001).

The analysis of patient satisfaction with the activities of nurses revealed that patients who regularly attend independent nursing consultations demonstrate a higher level of awareness, provide more favorable assessments of the quality of care, and report greater satisfaction compared to those who seek such services irregularly.

## 4. Discussion

The results of our study indicate that the main group of respondents consisted of women (63.9%) of middle and older age, with a median age of 62 years, a secondary level of education, and predominantly retired status. These patients are characterized by regular visits to the independent nursing consultation. At the same time, the working-age population seeks nursing care significantly less often. Comparison with international studies demonstrates similar trends. For example, according to a study based on data from the U.S. primary health care system [19], women constitute the overwhelming majority of patients consulting nurse practitioners, with a significant proportion belonging to the 60+ age category. These patients are more likely to have chronic conditions and require continuous monitoring, which explains their regular visits for care. Comparable results were reported in a Swiss study [20], where it was found that patients attending nurse practitioner consultations were predominantly elderly women with multimorbid conditions. The study highlights that these patients are more likely to require counseling, care, and regular follow-up, which explains their preference for nurse-led services [21].

The findings of our study also indicate that one of the most frequent reasons for patients visiting the independent nursing consultations obtaining subsidized medications (85.4% among informed patients and 68.2% among uninformed patients, *p* = 0.028). This trend is less frequently reported in international literature: in health care systems of the United States, Canada, Australia, and the United Kingdom, the predominant reasons for consulting nurse practitioners include chronic disease management, acute condition care, preventive examinations, and psychosocial support [22]. For instance, according to the study by Laurant et al. [23], in the Netherlands and the United Kingdom, up to 60% of nurse practitioner visits are associated with the management of chronic conditions such as diabetes, hypertension, and asthma, while prescribing medications is considered a secondary or complementary aspect rather than the main purpose of the consultation.

The overwhelming majority of respondents expressed positive opinions regarding the independent nursing consultations. Among those who attend regularly, 99.2% emphasized the necessity of this practice, and 98.3% noted the clarity of the recommendations provided. This reflects the high significance and utility of this model of health care delivery. Similar findings are reported in international research. For example, Julian Barratt [24] found that patients highly value interactions with nurse practitioners and express satisfaction with the information, care, and involvement in the treatment process. More than 90% of respondents stated that they would prefer to continue consultations with a nurse whenever possible.

Patient satisfaction in health care institutions continues to be a key indicator of the quality of medical services provided. Miriam Griffin emphasizes that patient satisfaction is an essential component in assessing the effectiveness of implementing new professional roles in health care systems. While there is evidence suggesting the positive impact of experienced nurse practitioners on clinical outcomes, it remains necessary to confirm these results at the national level, including regional and local data [25]. The findings of the present study support previous evidence of high patient satisfaction with advanced practice nursing services, particularly among older adults who require ongoing monitoring and preventive measures [26,27]. Regular visits to nurses contribute to improved health literacy, enhanced trust, and positive health outcomes, which is consistent with findings from Laurant’s research.

The high level of satisfaction with the independent nursing consultations among patients can be explained by the fact that the majority of respondents were retirees. This category of the population typically has an increased need for regular medical supervision, chronic disease management, and access to affordable health care services. Advanced practice nurses not only provide timely and convenient care, reducing waiting times, but also devote more attention to these patients by conducting extended consultations. Such an approach allows for a more detailed discussion of complaints, as well as clarification of preventive and treatment-related issues, thereby strengthening patient trust and satisfaction with health care services. Furthermore, older patients are more likely to demonstrate loyalty to medical staff, value stability, attentiveness, and personalized care—all of which contribute to the high evaluation of the services provided.

The findings of this study confirm the potential effectiveness of advanced nursing practice in Kazakhstan and underscore the need for its further expansion within primary healthcare settings. Sustainable development of this model requires strengthening the regulatory framework, clearly defining competencies, and standardizing the scope of practice for advanced practice nurses. An important practical direction involves the implementation of quality assessment tools, including structured checklists, to ensure consistency and accountability in service delivery. Increasing public awareness of the capabilities of nurse-led services may enhance service utilization and support a more efficient distribution of workload across healthcare professionals. Further research is needed to assess the impact of advanced nursing practice on clinical outcomes, cost-effectiveness, and the overall quality of healthcare services.

### Limitations

During the examination of the independent nursing consultation, several limitations were identified. In particular, it was found that the population remains largely oriented toward seeking care from physicians, which reduces the utilization of independent nursing consultation. According to the results of our study, the primary group attending the independent nursing consultation consisted predominantly of older adults, which limits the ability to obtain a fully objective and representative assessment of satisfaction with nursing consultation across the general population.

## 5. Conclusions

The findings of the study indicate a high level of patient satisfaction with the services provided by nurses during independent consultations. Most respondents gave positive assessments of both the organizational aspects of the visit and the quality of care, including clinical examination, monitoring of treatment dynamics, and clarity of the recommendations provided. In particular, 99.2% of patients with regular visits expressed satisfaction with the nurse’s work, while 76.6% rated her performance as “excellent”, which confirms the high level of trust in this model of care delivery.

At the same time, certain deficiencies were identified: among patients who rarely attended nurse-led consultations, there were more frequent reports of insufficient attention and knowledge on the part of the nurse, as well as lower awareness of the very existence of such a service.

## Figures and Tables

**Table 1 healthcare-13-03314-t001:** Inclusion and Exclusion Criteria for Patients.

Inclusion Criteria	Exclusion Criteria
Individuals aged 18 years and older, registered with the polyclinics, who attended an independent nursing consultation.	Patients younger than 18 years Individuals who declined to participate in the study Patients who had not attended the independent nursing consultation

**Table 2 healthcare-13-03314-t002:** Socio-demographic characteristics of respondents (n = 621).

Respondents		n	%
Gender	Male	220	35.4%
	Female	401	64.6%
Age	Me (Q1–Q3)	61.0 (51.3–68.0)	
Education	Incomplete secondary	34	5.7%
	Secondary	328	55.3%
	Secondary vocational	75	12.6%
	Higher	156	26.3%
Marital status	Married	448	75.7%
	Widowed	105	17.7%
	Divorced	23	3.9%
	Never married	16	2.7%
Social status	Worker	225	38.0%
	Employee	23	3.9%
	Student	3	0.5%
	Military personnel	1	0.2%
	Entrepreneur	17	2.9%
	Retiree	259	43.8%
	Homemaker	38	6.4%
	Unemployed	26	4.4%

**Table 3 healthcare-13-03314-t003:** Analysis of awareness indicators depending on the frequency of visits to nurse for independent consultations.

Parameters		How Often Do You Visit a Nurse for an Independent Consultation?				*p*
		Regularly		Rarely		
		n	%	n	%	
Are you aware of the concept of the independent nursing consultation?	No	19	3.6%	8	9.0%	0.021
	Yes	511	96.4%	81	91.0%	
From whom did you learn about an independent nursing consultation?	From physicians	258	48.6%	24	27.0%	0.001
	From nurses	280	52.7%	55	61.8%	0.112
	From relatives and friends	14	2.6%	8	9.0%	0.003
	From the call center	42	7.9%	7	7.9%	0.989
	From administrative staff at the registration desk	75	14.1%	10	11.2%	0.463

**Table 4 healthcare-13-03314-t004:** Analysis of indicators of organizational aspects depending on the frequency of visits to the independent nursing consultation.

Parameters		How Often Do You Visit a Nurse for an Independent Consultation?				*p*
		Regularly		Rarely		
		n	%	n	%	
What is the purpose of your visit to a nurse for an independent consultation	Preventive examination	303	57.1%	29	32.6%	<0.001
	Screening	267	50.3%	21	23.6%	<0.001
	Provision of subsidized medicines (prescription issuance)	442	83.2%	35	39.3%	<0.001
	Dispensary observation	384	72.3%	32	36.0%	<0.001
How did you sign up for an independent consultation?	Walk-in queue	409	77.0%	56	62.9%	<0.001
	By a nurse’s phone call	44	8.3%	20	22.5%	
	Through the Damumed/E-gov application	78	14.7%	13	14.6%	
How long did you wait for an independent consultation?	No more than 5–10 min	316	59.5%	63	70.8%	<0.022
	From 10 to 30 min	212	39.9%	24	27.0%	
	More than 1 h	3	0.6%	2	2.2%	
In your opinion, is an independent nursing consultation in outpatient clinics necessary?	Necessary	527	99.3%	84	94.4%	<0.002
	There is no need	4	0.8%	5	5.6%	

**Table 5 healthcare-13-03314-t005:** Analysis of indicators of quality depending on the frequency of visits to the independent nursing consultation.

Parameters		How Often Do You Visit a Nurse for an Independent Consultation?				*p*
		Regularly		Rarely		
		n	%	n	%	
Does the nurse conduct an examination and assessment of your general condition?	Yes	489	92.1%	77	86.5%	<0.188
	No	21	4.0%	5	5.6%	
	No answer	21	4.0%	7	7.9%	
If YES, does the nurse ask a sufficient number of questions to assess your overall condition?	Sufficient	517	97.4%	82	92.1%	<0.026
	Partially sufficient	10	1.9%	3	3.4%	
	Insufficient	2	0.4%	2	2.2%	
	Does not ask questions	2	0.4%	2	2.2%	
Does the nurse monitor the dynamics of your treatment?	Yes	525	98.9%	86	96.6%	<0.392
	No	6	1.1%	3	3.4%	
If yes, then in what way?	Provides repeated invitations for follow-up visits	456	85.9%	56	62.9%	<0.001
	Actively conducts home visits	47	8.9%	26	29.2%	<0.001
	Assesses the patient’s health status via telephone	69	13.0%	16	18.0%	<0.206

**Table 6 healthcare-13-03314-t006:** Analysis of indicators patient understanding of recommendations and changes in well-being depending on the frequency of visits to the independent nursing consultation.

Parameters		How Often Do You Visit a Nurse for an Independent Consultation?				*p*
		Regularly		Rarely		
		n	%	n	%	
Do you understand the nurse’s recommendations regarding the prevention and possible complications of your condition?	Yes	523	98.5%	83	93.3%	<0.003
	Not entirely	3	0.6%	4	4.5%	
	No recommendations were provided	5	0.9%	2	2.2%	
How has your well-being changed after the nursing consultation/intervention?	Improved	469	88.3%	73	82.0%	<0.040
	No change	61	11.5%	14	15.7%	
	Deteriorated	1	0.2%	2	2.2%	

**Table 7 healthcare-13-03314-t007:** Analysis of indicators of patient satisfaction depending on the frequency of visits to the independent nursing consultation.

Parameters		How Often Do You Visit a Nurse for an Independent Consultation?				*p*
		Regularly		Rarely		
		n	%	n	%	
Are you satisfied with the work of the nurse at the independent consultation?	Yes	527	99.2%	84	94.4%	<0.001
	Not entirely	4	0.8%	5	5.6%	
If you are dissatisfied with the work of the nurse conducting the independent consultation, please indicate the reason for your dissatisfaction.	Excessive hastiness in work	12	2.3%	6	6.7%	<0.001
	Insufficient attention to patients	2	0.4%	4	4.5%	
	Impolite communication with patients	2	0.4%	0	0.0%	
	Inability to establish rapport with patients	0	0.0%	1	1.1%	
	Insufficient professional knowledge	1	0.2%	3	3.4%	
Please evaluate the performance of the nurse conducting an independent consultation on a five-point scale.	2	1	0.2%	0	0.0%	<0.005
	3	4	0.8%	5	5.6%	
	4	119	22.4%	21	23.6%	
	5	407	76.6%	63	70.8%	

## Data Availability

The original contributions presented in this study are included in the article. Further inquiries can be directed to the corresponding author.

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
