# Peer review of "Patient Satisfaction with the Expanded Nurses Service in Primary Health Care: Evidence from Kazakhstan"

_healthcare, 2025, doi:10.3390/healthcare13243314_

Round 1

Reviewer 1 Report

Comments and Suggestions for Authors

The paper studies patient satisfaction with independent nurse-led consultations (advanced practice nursing, APN) in Kazakhstani primary care. Using a cross-sectional survey of 621 attendees across four facilities (April–Nov 2023), it compares “regular” vs “infrequent” users on awareness, organizational features, perceived quality, and satisfaction. It reports very high satisfaction overall, with systematically better ratings among regular attendees. This is a useful contribution as the literature tends to focus on perceptions of quality in health care delivery in advanced economies. There is far more limited evidence from Central Asia. 

However, beyond its geographic coverage, the paper falls short of really advancing our understanding of the drivers of perceptions of quality in health care. It is mostly descriptive (even though it often resorts to causal language), comparisons between regular and infrequent patients might be driven by selection (those who are satisfied return more frequently) and several empirical shortcomings are not adequately addressed (eg. social desirability bias, acquiescence bias, multiple hypotheses testing, etc). A revised version of the paper should also include more detailed descriptions of the scope of the tasks nurses perform, who conducted the surveys, etc. 

The paper is not yet ready for publication in this journal. 

Author Response

Dear Reviewer 1,

Thank you for taking the time to review this manuscript. We appreciate the opportunity to respond to the editors' and reviewers' comments and are pleased to take this opportunity to improve it. We have summarized the editors' and reviewers' comments and provided detailed responses to each point below.

We see that the article was significantly limited, and some points were clarified in response to the editor's constructive suggestions. Thank you for retaining this revised version for publication.

Sincerely,

Zhanar Dostanova

Comment 1: However, beyond its geographic coverage, the paper falls short of really advancing our understanding of the drivers of perceptions of quality in health care. It is mostly descriptive (even though it often resorts to causal language), comparisons between regular and infrequent patients might be driven by selection (those who are satisfied return more frequently) and several empirical shortcomings are not adequately addressed (eg. social desirability bias, acquiescence bias, multiple hypotheses testing, etc). A revised version of the paper should also include more detailed descriptions of the scope of the tasks nurses perform, who conducted the surveys, etc.

Response 1: Thank you for identifying this. We fully agree with your comments. To identify factors influencing patient satisfaction, we conducted a multivariate logistic regression analysis; however, no statistically significant differences were observed between the papers. This result underscores the descriptive nature of our study and, therefore, the need for larger, more detailed studies to better understand the contribution of the underlying factors. As this is the first study of its kind in Kazakhstan, advanced nursing practice has not yet been implemented in all healthcare organizations. It represents a critical initial step in exploring nurse-led independent conferencing models and in establishing an empirical foundation for future analytical and interventional studies. Comment 2: The revised version of the article should also include a more detailed description of the scope of tasks performed by nurses, interviewers, etc.

Thank you for your valuable comment. Changes within changes and highlighted in green. Lines 77–85; page 2.

Reviewer 2 Report

Comments and Suggestions for Authors

The abstract is structured; include key demographic data (e.g., "mostly older women") and provide actual p-values for the most significant findings (e.g., the difference in satisfaction between regular and rare visitors) to give a clearer picture of the strength of the evidence; check the keywords in accordance with MeSh.

In the introduction, please provide more examples of specific healthcare frameworks that assess patient matters, in relation to scientific literature (for e.g. doi: 10.3390/healthcare12030325). While the aim is clear, the introduction could more explicitly state the gap this study fills

The methods are well structured in subsections. Please provide the statistical software used with manufacturer and country.  The manuscript states the questionnaire was "pretested" and modified for validity and reliability; however, it should provide specific metrics from the pilot testing, such as Cronbach's alpha for internal consistency of the domains. The "rarely" category is vague ("infrequently or occasionally")- providing a more specific definition (e.g., "less than once every three months") would enhance clarity.

The results are comprehensive and presented logically with clear tables. While the data is presented, the narrative could do more to synthesize and emphasize the most critical results. For example, explicitly state that regular visitors were over 10 times more likely to report "excellent" satisfaction and had significantly higher rates of perceived health improvement.

The discussion effectively interprets the main findings, correctly linking the high satisfaction among regular (and predominantly older) visitors to their need for continuous care.

The conclusion accurately summarizes the main finding of high satisfaction and correctly identifies the key challenges

The references are adequate but few given the type of paper.

Author Response

Dear Reviewer 2,

Re: Manuscript Patient Satisfaction with the Expanded Nurses Service in
Primary Health Care: Evidence from Kazakhstan.

Thank you very much for taking the time to review this manuscript. We appreciate the opportunity to respond to the editors’ and reviewers’ comments on our manuscript, and we are pleased to use this chance to improve it. Below, we have summarized the editors’ and reviewers’ comments and provided our detailed responses to each point.

We believe that the paper has now been much improved, and the key messages clarified in response to the constructive suggestions of the editor.  Thank you for considering this revised version for publication.

Yours sincerely,

Dostanova Zhanar

Comments 1: The abstract is structured; include key demographic data (e.g., "mostly older women") and provide actual p-values for the most significant findings (e.g., the difference in satisfaction between regular and rare visitors) to give a clearer picture of the strength of the evidence; check the keywords in accordance with MeSh.

Response 1: Thank you for pointing this out. I have verified all four keywords against the MeSH terminology, and they all correspond to official MeSH descriptors.

Comments 2: In the introduction, please provide more examples of specific healthcare frameworks that assess patient matters, in relation to scientific literature (for e.g. doi: 10.3390/healthcare12030325). While the aim is clear, the introduction could more explicitly state the gap this study fills.

Response 2: Thank you for pointing this out. Changes have been made and highlighted in green. Line 58 – 73; page 2, Line 101 – 106; page 3.

Comments 3: The methods are well structured in subsections. Please provide the statistical software used with manufacturer and country.  The manuscript states the questionnaire was "pretested" and modified for validity and reliability; however, it should provide specific metrics from the pilot testing, such as Cronbach's alpha for internal consistency of the domains. The "rarely" category is vague ("infrequently or occasionally")- providing a more specific definition (e.g., "less than once every three months") would enhance clarity.

Response 3: Thank you for your note. Changes have been made and highlighted in green.

Line 157 – 159; page 4, Line 121 – 124; page 3, Line 167 – 172; page 4.

Comments 4: The references are adequate but few given the type of paper.

Response 4: Thank you for your note. We have added 3 references and highlighted in green.

Line 482 – 487; page 14-15.

Reviewer 3 Report

Comments and Suggestions for Authors

The manuscript is very well structured and written, and the topic it brings is very current. The title of the paper is well conceived and fully related to the content. Overall, the manuscript has many strengths and very few weaknesses.

The introduction is well conceived. In this section, the context and goal of the research is clearly defined. The description of "advanced practice nurse" applied in Kazakhstan seems too generic. This part needs to be given more attention with a clear description and specifics of this practice in the mentioned country.

The sampling procedure and research implementation is not completely clear. How were the respondents approached? Who conducted the research? When was the survey conducted: immediately after the medical treatment or later? How is objectivity and impartiality ensured in the collection of responses? These are important questions that need to be addressed, while also strengthening the validity and credibility of the study.

A manuscript remains without purpose if there are no practical implications. Essentially, the results of the study are related to the pilot project of introducing advanced nursing practice in a limited number of healthcare organizations. What are the conclusions and further practical implications regarding advanced nursing practice? This missing element of the manuscript is probably easiest to integrate into the Discussion section.

What limitations does the study have? Potential limitations of the study are at the same time guidelines for further development of the Discussion section. For example, are the effects of greater patient satisfaction with the Expanded Nurses Service reflected in increased overall satisfaction with the service provided? Is it an isolated effect or does it contribute to the overall increase in the quality of the service provided, and indirectly to the overall satisfaction of patients? The influence of the following factors should also be considered: extended communication with patients, time allocated for medical treatment and conversation with the patient, greater perceived commitment of nurses, development of social relationships, and the like. This needs to be analyzed and discussed. Also, the question of whether the Expanded Nurses Service contributes to increasing patient satisfaction or contributes to the improvement of their treatment is fundamentally opened. Is there a specific "quasi-placebo effect" in this approach, or is there a fundamental improvement in medical practice from which patients benefit? These are open questions that should indicate the real effects of the Expanded Nurses Service.

The manuscript is very inspiring and of importance to a wide audience.

Author Response

Dear Reviewer 3,

Re: Manuscript Patient Satisfaction with the Expanded Nurses Service in
Primary Health Care: Evidence from Kazakhstan.

Thank you very much for taking the time to review this manuscript. We appreciate the opportunity to respond to the editors’ and reviewers’ comments on our manuscript, and we are pleased to use this chance to improve it. Below, we have summarized the editors’ and reviewers’ comments and provided our detailed responses to each point.

We believe that the paper has now been much improved, and the key messages clarified in response to the constructive suggestions of the editor.  Thank you for considering this revised version for publication.

Yours sincerely,

Dostanova Zhanar

Comments 1: The introduction is well conceived. In this section, the context and goal of the research is clearly defined. The description of "advanced practice nurse" applied in Kazakhstan seems too generic. This part needs to be given more attention with a clear description and specifics of this practice in the mentioned country.

Response 1: Thank you for pointing this out. Changes have been made and highlighted in green. Line 77 – 85; page 2.

Comments 2: The sampling procedure and research implementation is not completely clear. How were the respondents approached? Who conducted the research? When was the survey conducted: immediately after the medical treatment or later? How is objectivity and impartiality ensured in the collection of responses? These are important questions that need to be addressed, while also strengthening the validity and credibility of the study.

Response 2: Thank you for pointing this out. Changes have been made and highlighted in green.  Line 135 – 144; page 3-4.

Comments 3: A manuscript remains without purpose if there are no practical implications. Essentially, the results of the study are related to the pilot project of introducing advanced nursing practice in a limited number of healthcare organizations. What are the conclusions and further practical implications regarding advanced nursing practice? This missing element of the manuscript is probably easiest to integrate into the Discussion section.

Response 3: Thank you for your note. Changes have been made and highlighted in green. Line 362 – 372; page 12.

Comments 4: What limitations does the study have? 

Response 4: Thank you for pointing this out. Changes have been made and highlighted in green. Line 374 – 380; page 12.

Comments 5: Potential limitations of the study are at the same time guidelines for further development of the Discussion section. For example, are the effects of greater patient satisfaction with the Expanded Nurses Service reflected in increased overall satisfaction with the service provided? Is it an isolated effect or does it contribute to the overall increase in the quality of the service provided, and indirectly to the overall satisfaction of patients? The influence of the following factors should also be considered: extended communication with patients, time allocated for medical treatment and conversation with the patient, greater perceived commitment of nurses, development of social relationships, and the like. This needs to be analyzed and discussed. Also, the question of whether the Expanded Nurses Service contributes to increasing patient satisfaction or contributes to the improvement of their treatment is fundamentally opened. Is there a specific "quasi-placebo effect" in this approach, or is there a fundamental improvement in medical practice from which patients benefit? These are open questions that should indicate the real effects of the Expanded Nurses Service.

Response 5: Thank you for your comments.

The findings of this study indicate that patient satisfaction with services provided by advanced practice nurses (APNs) may contribute to overall satisfaction with primary health care services. However, this influence cannot be considered isolated within the context of our research. Overall patient satisfaction is shaped by a wide range of factors, including the accessibility of physician services, organizational processes, and patient expectations. Therefore, determining the exact contribution of APN-led consultations requires further investigation.

Satisfaction with APN services is likely not an isolated effect. It may be mediated by improvements in nurse–patient interactions, greater accessibility of consultations, and a more patient-centred approach. This suggests a comprehensive enhancement of perceived service quality rather than a distinct effect attributable solely to the APN role.

Patient satisfaction may also be influenced not only by the competencies of the APN but by accompanying factors such as consultation length, individualized care, provider commitment, and the development of trusting relationships. Although these factors were not directly assessed in our study, international evidence demonstrates their significant contribution to positive perceptions of nursing services. Future research should incorporate these variables into study designs.

Our study does not allow for a definitive conclusion regarding whether advanced nursing practice leads to improved clinical outcomes. Satisfaction represents a subjective perception of service quality, whereas clinical outcomes require objective assessment, including changes in health status, chronic disease control, and preventive effects. Further studies should include clinical indicators to determine the true contribution of APN practice to treatment effectiveness.

The possibility of a so-called “quasi-placebo effect,” whereby patients respond positively to the new format of care due to increased attention and the novelty of the approach, cannot be excluded. However, the sustained nature of patient satisfaction and evidence from international studies suggest that advanced nursing practice also yields structural improvements—enhanced accessibility, continuity, and personalization of care. Longitudinal studies are needed to differentiate these effects and assess the durability of observed changes over time.

Demonstrating the true impact of advanced nursing practice requires a comprehensive assessment that includes patient satisfaction, clinical outcomes, accessibility indicators, and the effectiveness of care coordination. This study represents an initial step toward that goal by capturing patients’ subjective evaluations and providing a foundation for subsequent research focused on objective indicators of effectiveness.

Round 2

Reviewer 1 Report

Comments and Suggestions for Authors

The author has addressed my concerns.